# Epigenetic Clock and Circadian Rhythms in Stem Cell Aging and Rejuvenation

**DOI:** 10.3390/jpm11111050

**Published:** 2021-10-20

**Authors:** Ekaterina M. Samoilova, Vladimir V. Belopasov, Evgenia V. Ekusheva, Chao Zhang, Alexander V. Troitskiy, Vladimir P. Baklaushev

**Affiliations:** 1Federal Research and Clinical Center of Specialized Medical Care and Medical Technologies, FMBA of Russia, 115682 Moscow, Russia; info@fnkc-fmba.ru (A.V.T.); serpoff@gmail.com (V.P.B.); 2Neurology Department, Astrakhan State Medical Academy, 414000 Astrakhan, Russia; belopasov@yandex.ru; 3Academy of Postgraduate Education of the Federal Scientific and Clinical Center for Specialized Types of Medical Care and Medical Technologies, FMBA of Russia, 125371 Moscow, Russia; ekushevaev@mail.ru; 4Tianjin’s Clinical Research Center for Cancer, Tianjin 300060, China; drzhangchao@tmu.edu.cn

**Keywords:** circadian clock, epigenetic clock, aging, senescence, reprogramming, methylome, DNA methylation, CpG-islands, induced pluripotent stem cells

## Abstract

This review summarizes the current understanding of the interaction between circadian rhythms of gene expression and epigenetic clocks characterized by the specific profile of DNA methylation in CpG-islands which mirror the senescence of all somatic cells and stem cells in particular. Basic mechanisms of regulation for circadian genes CLOCK-BMAL1 as well as downstream clock-controlled genes (ССG) are also discussed here. It has been shown that circadian rhythms operate by the finely tuned regulation of transcription and rely on various epigenetic mechanisms including the activation of enhancers/suppressors, acetylation/deacetylation of histones and other proteins as well as DNA methylation. Overall, up to 20% of all genes expressed by the cell are subject to expression oscillations associated with circadian rhythms. Additionally included in the review is a brief list of genes involved in the regulation of circadian rhythms, along with genes important for cell aging, and oncogenesis. Eliminating some of them (for example, *Sirt1*) accelerates the aging process, while the overexpression of *Sirt1*, on the contrary, protects against age-related changes. Circadian regulators control a number of genes that activate the cell cycle (*Wee1*, *c-Myc*, *p20*, *p21*, and *Cyclin D1*) and regulate histone modification and DNA methylation. Approaches for determining the epigenetic age from methylation profiles across CpG islands in individual cells are described. DNA methylation, which characterizes the function of the epigenetic clock, appears to link together such key biological processes as regeneration and functioning of stem cells, aging and malignant transformation. Finally, the main features of adult stem cell aging in stem cell niches and current possibilities for modulating the epigenetic clock and stem cells rejuvenation as part of antiaging therapy are discussed.

## 1. Introduction

The aging process is characterized by a progressive decline in multiple physiological functions. It is accompanied with reduced metabolic activity, increase in adipose tissue, loss of muscle mass and function [1,2], de-regulation of circadian rhythms controlling sleep/wake cycles, weaker immunity [3,4], impaired cognitive functions, retinal dysfunction, etc. [5,6]. Just as any biological process, aging is the result of a complex interaction between the genetic programs of an organism and environmental factors. Endogenous as well as external aging factors operate through such mechanisms as circadian rhythms and epigenetic clocks. Circadian rhythms, which are diurnal fluctuations in genetic expression, have been known about for a long time, and their physiological influence can hardly be overestimated. The circadian clock, synchronizing with external signals, sets the rhythm for almost all physiological parameters from cognitive functions, caused by the level of activity of neurotransmitter systems, to fluctuations of metabolic parameters in each individual cell. [7,8,9,10]. It is well known that frequent movement across time zones leading to a mismatch between the outer day/night cycle and the inner clock is a powerful stressor, increasing the risk of neurotic disorders, cardiovascular disease, obesity, metabolic disorders and cancer. [11,12,13]. However, in our opinion, the role of circadian clocks in aging and cell senescence remains underappreciated.

The concept of epigenetic clocks has been proposed quite recently, and began with the works of Hannum G. [14] and Horvath S. [15], who in 2013 showed that DNA methylation profile is one of the most accurate markers of cell aging. Subsequent studies established the existence of a genetically pre-determined multi-level program downstream of CpG methylation [16,17,18], which defines the epigenetic age of individual cells and of the entire organism.

The relationship between circadian rhythms and the epigenetic clock and their combined effect on the cell cycle is of great research interest. At the moment, such a relationship is shown for cell cycle defects accompanying oncogenic transformation [19]. Molecular mechanisms of epigenetic regulation of the circadian rhythms of transcription and their role in the regulation of the cell cycle during aging are currently being actively studied [20,21,22].

The purpose of this review is to summarize the currently available data on the molecular organization of the two main clock mechanisms (pacemakers) of the body, their mutual regulation and influence on stem cell aging, as well as to outline potential approaches to the management of epigenetic aging processes in the framework of antiaging therapy.

## 2. Circadian Clocks and Cell Aging

In the process of evolution, all life forms on Earth except for the minority of deep sea and underground species have adapted to a 24-hr day/night cycle, and their biological processes became subject to cyclical fluctuations known as the circadian clock (from Latin circa diem around a day). Diurnal cycles contribute to a vast array of physiological and metabolic functions, such as sleep/wake rhythms, cognitive activity, blood pressure changes, hormone production, etc. [23]. In mamalians, central circadian pacemaker responsible for tuning to the daily cycle is found in the suprachiasmatic nucleus (SCN) It undergoes light-induced daily activation via retinohypothalamic tract (RTH). In addition, non-photic stimuli, such as food cues, including meal times and dietary composition, may also contribute to the activation of the central pacemaker [24,25,26,27]. In addition to the SCN, peripheral clocks are located in other areas of the brain, including nuclei in the hypothalamus and hippocampus, as well as in peripheral tissues, i.e., the liver, kidneys, heart, etc. They are synchronized with the central clock, but are also influenced by additional stimuli, such as food intake, hormone secretion, etc.

Recent studies have demonstrated that age-related changes in circadian physiology are also detectable at the cellular level. In vivo multiunit neural activity recordings from the SCN have uncovered age-related decline in the amplitudes of electrical activity rhythms [28]. Electric properties of neurons in the SCN change during aging, with older neurons displaying attenuated circadian control of potassium channel activity and compromised physiological properties of the cell membrane [29]. These electrophysiological changes may result from disrupted function of calcium-activated potassium channels [30]. The quantification of synaptic terminals in the SCN demonstrated age-related reductions in the number of synaptic spines and shortening of dendrites [31]. In addition, there is an age-related decrease in the expression of such neuropeptides as vasoactive intestinal peptide, (VIP) [32] and arginine-vasopressin peptide (AVP) in SCN [33]. The neuropeptides VIP and AVP play an important role in the synchronization of cellular rhythms in the SCN [34]; therefore, a decrease in their expression inevitably disrupts the function of the central circadian clock.

### 2.1. Transcriptional and Metabolic Mechanisms of Circadian Rhythms

Molecular mechanisms underlying the circadian rhythms are organized into a complex hierarchical network of transcription-translation feedback loops [10] (Table 1). Central proteins of the mammalian clockworks are CLOCK (Circadian Locomotor Output Cycles Kaput) and BMAL1 (Brain and Muscle ARNT-Like 1). They serve as the major transcription activators of b-HLH-PAS (The basic helix–loop–helix/Per-ARNT-SIM) expression, which in turn modulate the transcription of multiple clock-controlled genes (CCGs). BMAL1proteins heterodimerize through their PAS domains and bind the E-box enhancer elements in the promoter regions of CCGs. Two genes under the CLOCK/BMAL1 control, i.e., Period1-2 (Per1-2) and Cryptochrome1-2 (Cry1-2), encode the main clock driving proteins PER and CRY, which, in turn, heterodimerize and bind other co-factors to repress the activity of CLOCK-BMAL1. Thus, a negative feedback loop is formed [35]. A number of CCGs encode transcription factors, such as DBP, TEF, RORα, and REV-ERBα/β). DBP and TEF associate with D-boxes, whereas RORα and REV-ERBα/β occupy the promoter elements of Reb-Erb/ROR, thereby setting up additional circadian waves in transcription of downstream genes.

Temporal and tissue specificity of CCGs controls the activity of peripheral pacemakers throughout the body. Many genes immediately downstream of the pacemaker activity are transcription factors that drive secondary transcription oscillations in the cells and produce the rhythms that are uncoupled from the direct control of the central pacemaker. Detailed expression profiling has shown that rhythms initiated by distinct transcription factors may differ in their properties (for instance, DBP-based regulation is characterized with higher-amplitude rhythmic activity, compared to that of the E-box driven regulation) [36]. Additionally, a number of post-translational mechanisms such as RNA interference [37] and protein ubiquitination [38] may add an extra layer of regulation to the cyclic control by central pacemaker, CCGs and downstream gene networks. These multiple levels of regulation provide dynamic control of central and peripheral pacemakers. According to some estimates, about 10–20% of genes in each cell are under the control of circadian clocks and display cyclic oscillations in expression [39]. Furthermore, a recent report indicates that many more genes may display cyclic expression once the nutritional and metabolic factors are taken into account [40].

Epigenetic control plays a central role in harmonizing the circadian oscillations in transcription, and is mediated primarily via chromatin remodeling [41]. CLOCK is known to have a HAT activity, preferring H3K9 and H3K14 as the substrates [42]. This activity likely promotes the hetero- to euchromatin transition and stimulates the cyclic activation of CCGs [43]. A number of HATs, including CBP and p300, are dynamically recruited by the CLOCK-BMAL1 complex, with p300 apparently functioning as the co-activator of CLOCK-BMAL1 and launching the cyclic acetylation of histone tails [44,45]. Less is known about the role of CBP, but it may operate similarly to p300, and has been described to interact with PER2 to repress CLOCK-BMAL1 activity [44]. Additionally, CLOCK-BMAL1 is known to recruit MLL1 so that it tri-methylates histone H3K4 on the nucleosomes near the circadian gene promoters, which results in transcription activation [46,47]. On the contrary, EZH2 methylates H3K27 on CCG promoters thereby resulting in CRY-dependent suppression of transcription [48]. Histone demethylases Jarid1A and Jmjd5 have also been implicated in the functioning of circadian epigenome [47,49]. Circadian acetylation of H3K9 and H3K14 residues on the nucleosomes found at CCG promoters as well as of the circadian nuclear proteins BMAL1 and PER2 is mediated by the NAD ^+^ -dependent deacetylases SIRT1 [50,51] and SIRT6 [52].

In humans and mice, seven Sirtuin family members are present, and are characterized with a conserved catalytic domain and variable N- and C-terminal extension regions (Table 1). SIRT1, SIRT2, SIRT3, SIRT6, and SIRT7 display pronounced protein deacetylase activity, unlike SIRT4 and SIRT5 that are lacking it, but are capable of removing other acyl (rather than acetyl) groups attached to lysine residues in the proteins [53]. Notably, SIRT1, SIRT2, SIRT6, and SIRT7 apparently function as epigenetic regulators, whereas SIRT3, SIRT4, and SIRT5 locate to the mitochondria [53]. Mammalian sirtuins SIRT1 and SIRT6 have been demonstrated to control circadian rhythms [51,52] as well as metabolism, aging, and cancer. The exact role of SIRT1 in cancer is presently not well understood: SIRT1 was found to deacetylate a number of proteins downstream of MYC, P53, HIF, TGF-β, and WNT signaling, and acts as a tumor suppressor or a tumor-promoting factor, depending on the specific biological context [54]. Additionally, SIRT1 is involved in the adaptation of the organism to nutritional starvation. Upon calorie restriction, it launches lipid mobilization from adipose tissue, switches skeletal muscles and liver from glucose to lipid oxidation, and boosts hepatic glucose production [55]. SIRT1 is required for VMH and ARC neurons to control glucose metabolism and lipid exchange in the periphery; deregulation of these processes is associated with aging [55,56]. It is important to note that, in the context of SCN, SIRT1 modulates the activity of the central pacemaker, and this modulation, mediated by BMAL1 deacetylation via the NAMPT/SIRT1/PGC-1α loop, likely becomes weaker as the organism ages [57,58].

One more important factor bridging together circadian clocks and aging is the control of energy metabolism in the mitochondria. In-depth analysis of circadian acetylome has demonstrated that most of the acetylation events downstream of the circadian clocks are related to the mitochondrial proteins [59]. This observation is consistent with clock-modulating activity of SIRT3 known to target mitochondrial proteins. SIRT3 is believed to set the acetylation rhythms as well as OXPHOS oscillations in liver mitochondria [60]. Furthermore, upon ROS, the cells may reset their circadian clocks by activating antioxidant pathways via BMAL1, HSF1, and CK2 [61]. SIRT1 also contributes to the mitochondrial functioning by modulating the nuclear-mitochondrial communication and thereby preventing aging [62]. On the other hand, SIRT1 is itself under the control of circadian clocks. The synthesis of NAD^+^ is directly controlled via CLOCK-BMAL1 regulation of *NAMPT* gene encoding NAMPT, which catalyzes the key step in NAD ^+^ biogenesis. Pharmacological or genetic inhibition of NAMPT activity decreases intracellular NAD ^+^ levels and SIRT1 activity, resulting in the restored acetylation of H3K9 and H3K14 on CCG promoters, as well as in hyperacetylation of BMAL1, which in turn suppresses the expression of CCGs [51]. For instance, in old mice, the levels of SIRT1 in SCN were reduced, and young *sirt1*-knockouts display a premature aging phenotype, which includes the elongated circadian period and down-regulation of Per2 and BMAL1. *Sirt1*-overexpressing mice, in turn, were found to be largely protected against these age-dependent changes [57].

### 2.2. Cell Cycle Control

Circadian clocks contribute to the control of G1/S [63] and G2/M [64] transition by synchronizing the two rhythms together [65]. Circadian clocks have been observed to control expression of a number of important cell cycle regulator genes such as *Wee1*, *c-Myc*, *p20*, *p21* and *Cyclin D1* [66,67]. Notably, key circadian regulators linking the clocks to the cell cycle, are PER1 and PER2. Several cell-cycle genes remain silent in *per2*-mutant mice, whereas PER1 and PER2 interact with checkpoint control kinases CHK1/2 [68] and Nono/p16-Ink4A, and contribute to the fine-tuned control of cell cycle [69].

De-regulation of circadian rhythms has been increasingly associated with the pathogenesis of multiple diseases including cancer. Two of the most prominent examples of cell cycle regulators associated with circadian clocks include tumor suppressor *p53* and an oncogene *c-Myc*. *p53* has been demonstrated to control several cell cycle checkpoints [70], and regulates *per2* expression via blocking the association of CLOCK-BMAL1 with *per2* promoter [71]. Mice with *tp53* knockout were found to have a shorter circadian period [71]. *c-Myc* is one of the key cell cycle regulators [72,73], and binds E boxes to control transcription response, similarly to CLOCK-BMAL1. Thus, the circadian networks are extensively intertwined with the oncogenic signaling pathways and mediate the fine-tuned control of cell cycle. De-regulation of this control likely contributes to the cell malignant transformation.

## 3. Epigenetic Clocks

Changes in CpG methylation profiles are now accepted as a canonic epigenetic feature of aging [74,75]. Total DNA methylation is known to decrease with age, yet the CpG islands, particularly those found in Polycomb-target genes become hypermethylated [18,76]. Several OMIX approaches in the whole-genome single-cell sequencing mode help analyze the methylome in individual cells or tissues, and thereby obtain a more detailed understanding of age-dependent dynamics of CpG methylation [15,77,78].

The process of DNA methylation and demethylation is closely interrelated with the modification of histones, in particular with the methylation of lysine in their composition. Lysine methylation may be associated with DNA methylation as well as demethylation. For instance, H3K4me3 is associated with gene promoter demethylation, whereas H3K36me3 and H3K9me3 are responsible for hypermethylation [79]. The opposite is also true: DNA hypomethylation results in the redistribution of Pc-G proteins and H3K27me3 [80,81].

Epigenetic clocks are a regression model linking changes in methylation of specific CpGs with biological age [14,82]. Over the past decade, prognostic models based on the DNA methylation levels have relied on machine learning algorithms, and this technology has virtually revolutionized the research area pertaining to aging. Epigenetic clocks based on the DNA methylation are currently a better estimate of actual chronological age than any other transcriptome and proteomic data, including those based on the telomere length analysis [16]. Originally created for the assessment of chronological age, epigenetic clocks now integrate and predict various parameters of biological aging and disease risks, thereby establishing their clinical relevance [16].

The first epigenetic clocks have been developed by Hannum et al. based on the methylation status of 71 CpGs from PBMCs [14]. The next version of pan-tissue epigenetic clocks, which was in many ways much more accurate, was reported by S. Horvath based on the methylation of 353 CpGs in genomic DNA isolated from several normal tissues and tumors [15]. Recently developed pan-tissue mammalian epigenenetic clocks can provide an estimate of epigenetic age for virtually any mammalian tissue with an impressive accuracy [83]. Nonetheless, pan-tissue epigenetic clocks inform us of the chronological age and the rate of aging regardless of the contribution of external factors and tissue specificity.

True biological age is affected by several intrinsic and extrinsic factors. For this reason, the most advanced epigenetic clocks are either tissue-specific, or take into account additional parameters besides the DNA methylation profile (for example, PhenoAge, GrimAge) [74,84]. In particular, the “GrimAge” clock, in addition to analyzing the methylation profile, includes the determination of plasminogen activator inhibitor 1 (PAI-1) and growth differentiation factor 15 in the blood serum, taking into account smoking and comorbidity, which contributes to a more accurate prediction of life expectancy and quality of life [84]. Epigenetic clocks are of particular interest for the regenerative medicine, as these models are sensitive enough to detect even minor changes in the biological age resulting from longevity and reprogramming interventions [85,86].

The majority of epigenetic clocks uses the information from samples composed of multiple cells as an input [87], which results in the loss of their epigenetic heterogeneity [87,88]. Single-cell formats of methylome profiling have addressed this issue, and now include scRRBS and scWGBS/scBS pipelines [88,89,90].

Recently, a new method for the analysis of DNAm scAge profiles has been developed [91], making it possible to determine the epigenetic age of individual cells based on a probabilistic algorithm, largely independent of which CpGs are covered in each specific cell.

This method reproduces the chronological age of the tissue on average and also reveals the intrinsic epigenetic heterogeneity that exists between cells. As a result, the authors found that the combination of several predictions for individual cells gives an accurate average of the age of a particular tissue. These results suggest a high heterogeneity of the aging process even within one tissue, depending on many epigenetic factors, type and biological characteristics of cells (stem, proliferating, terminally differentiated, etc.) The authors of the study also suggested that some cells undergo accelerated or delayed epigenetic aging, which was previously impossible to establish [91]. The data obtained allow us to conclude that in the process of aging of cells and tissues, the clock probably “ticks” independently within each cell.

## 4. Are Circadian and Epigenetic Clocks Interrelated/Interconnected?

The existence of two universal, but inherently different clock mechanisms dynamically controlling most biological events at the organismic as well as at cellular levels raises the question of whether the circadian and epigenetic clocks interact with each other. This interaction is indirectly evidenced by, for example, evidence of dynamic DNA methylation in the central clock region of the SCN. Researchers have identified light-induced changes in DNA methylation of certain promoters that correspond to the expression of circadian genes [92]. Vice versa, global changes in DNA methylation, which occur, for example, in cancer cells, are accompanied with altered promoter methylation of circadian genes *per* and *cry* [93,94]. Furthermore, DNA methyltransferases and histone-modifying enzymes such as various HDACs, HMTs, and histone demethylases are known to interact with as well as control the recruitment of each other, suggesting a complex interaction between the factors involved in epigenetic control [95]. It is noteworthy that the primary cells isolated from the tissue lose their native rhythm in relation to both the circadian and epigenetic clocks [96].

Lehmann M. et al. [97] report that massive synchronized changes in methylation patterns and the aging rate of normal tissue in mammals occur three times in certain periods, i.e., before the age of one year, during puberty, and after the age of twenty. It is difficult to imagine that cells are able to synchronously change their methylation patterns over such long periods of time without the presence of a common synchronizer in the system, the role of which is quite suitable for a circadian clock with a central oscillator in the SCN. However, there is no direct evidence to date of the influence of circadian rhythms on methylation of CpG DNA islands.

## 5. Managing the Process of Cellular Aging. Can We Fool the Clock?

Preserving youth and prolonging active life has always been relevant for humanity. The development of the epigenetic clock concept has made it possible to study aging processes at a new level, for individual cells and tissues as well as for the organism as a whole. It was found that certain therapeutic effects can change the readings of the epigenetic clock in the direction of decreasing biological age (while keeping the same chronological age) and increasing the predicted life expectancy. In particular, TRIIM (Thymus Regeneration, Immunorestoration, and Insulin Mitigation) study showed that treating healthy males aged 51–65 years for several months with a combination of recombinant human growth hormone (rhGH), dehydroepiandrosterone (DHEA) and metformin helped reverse the immunosenescent trends, which was accompanied with a significant decrease in the epigenetic age as assayed by GrimAge [98].

A recent study on a rat model system [99] used six different epigenetic clocks based on the DNA methylation. It demonstrated that infusing plasma components from young to old rats for 5 months resulted in the normalization of biochemistry of old animals approaching that of the young animals. Most interestingly, the epigenetic age of blood, liver and heart in such animals turned out to be almost three times lower (25 months to 7 months reversion). Other markers of cell aging unrelated to the epigenetic clocks were similarly reduced, which was indicative of animal rejuvenation.

These very impressive results nonetheless leave open the questions of whether such approaches may affect the epigenetic age of the entire organism and whether the effects are reversible, once the pharmacological support has been discontinued. This issue is especially relevant given the pronounced heterogeneity of aging processes in various tissues, which is largely due to the peculiarities of the functioning of resident stem cells. The presence of stem cell niches determines the regenerative potential of organs and tissues, and their functional state undoubtedly affects epigenetic age. In this regard, the aging of stem cells as well as approaches to their “rejuvenation” require separate consideration.

### 5.1. Resident Stem Cells and Aging

Regeneration of almost all organs and tissues is carried out with participation of resident stem cells (SC), which are characterized by the ability to self-renew and differentiate into various cell types due to asymmetric division. Initially, it was believed that SCs are not subject to replicative aging, but nowadays there is enough evidence that they, like other cells, accumulate metabolic and genetic damage with age and are exposed to age-related epigenetic factors [100]. In addition, the criterion for the aging of organs and tissues is a decrease in the proportion of stem cells in the corresponding niches [100]. Let us take a closer look at the aging of the main types of stem cells.

**Hematopoietic stem cells (HSCs)** are found in the bone marrow of adult mammals, and are responsible for hematopoiesis. In the process of HSC aging, their clonal diversity decreases, due to a decrease in the intensity of proliferation of individual clones, which leads to a depletion of the clonal composition of all subsequent cell generations [101] Despite the fact that the total proportion of HSC does not change or even increases, the depletion of the clonal composition indicates a decrease in the number of functionally active HSC clones [102,103,104]. Another defining characteristic of HSC aging is a shift in the profile of their differentiation towards the myeloid row due to a decrease in the lymphoid row [102,105]. These data are in good agreement with the abovementioned observations of immune senescence, which is accompanied by weaker adaptive cell immunity due to the shrinkage of TCR repertoire diversity [98]. Age-related changes affecting all links of the lymphoid lineage and a shift in differentiation towards myelogenesis determine, in particular, a higher frequency of myeloid leukemia in older persons. [106,107]; these changes are probably the reason for the age-related growth of oncological pathology in general.

**Multipotent mesenchymal stem cells (MSCs)** are found in the bone marrow, adipose tissue as well as in several other organs and tissues of the human body. MSCs differentiate into multiple cell types such as fibroblasts, bone, cartilage, fat, muscle and other stromal cells, and play an important role in the regeneration of the above tissues [108,109]. Functionally, the aging of MSCs is characterized by a decrease in their proliferative activity and, as a consequence, a decrease in their proportion in the bone marrow and other tissues, as well as a decrease in their ability to differentiate. [110]. Aging MSCs display granular morphology and change the profile of secreting molecules. This phenomenon is referred to as senescence-associated secretory phenotype (SASP) [111]. In addition, aging of MSCs is accompanied by changes in nuclear morphology and the formation of a distinct chromatin structure called aging-associated heterochromatic foci (SAHF) [112]. Currently, MSC aging is routinely assessed by measuring beta-galactosidase activity, telomere length, gene expression markers, gene methylation, and other epigenetic markers [113].

**Intestinal Stem Cells (ISCs)** support the regular renewal of the gastrointestinal tract epithelium. Most of our knowledge of ISC aging comes from studies in Drosophila, where ISCs are easily identified by the expression of the snail transcription factor (Esg) and the Notch delta ligand (D1). During aging, the number of cells with the ISC immunophenotype increases several times, which is accompanied by a decrease in their function. [114,115]. This increase is associated with a disruption of the self-renewal process, accompanied by partial differentiation of ISCs with the preservation of stem cell markers [114]. In mammals, there are two interconverting populations of ISCs: proliferatively active Lgr5-expressing cells located at the base of the crypt and resting stem cells at several positions above the base of the crypt [116]. Before these markers were known, radiation experiments showed that although the GIT becomes more susceptible to damage with age, the total number of clone-forming units increases. [117]. It is assumed that aging of human ISC with impaired self-renewal and differentiation contributes to an increase in the incidence of colorectal cancer with age. [118].

**Satellite stem cells (myosatellite cells)** assure the regeneration of damaged skeletal muscles [119,120]. Unlike HSCs and ISCs, the number of satellite cells decreases markedly with age [121,122]. The in vitro proliferation rate and the potential for engraftment and regeneration of satellite cells after in vivo transplantation also decrease with age [123,124,125]. Moreover, just like HSCs, old satellite cells show a distorted differentiation potential, causing them to differentiate towards fibroblasts rather than myoblasts, mainly due to changes in Wnt and Tgf-β signaling [126,127]. Heterochronous transplantation of satellite cells from aged to young mice indicates that the mechanisms underlying changes in the regeneration potential of satellite cells include changes in the availability of ligands of the Wnt, Notch, Fgf, and Tgf-β superfamily [126,128,129], as well as changes in cytokine signaling through the JAK-STAT pathway [130]. In addition to changes in the microenvironment, self-renewal defects, and enhancement of stress-induced p38-MAPK signaling are associated with aging of the satellite cells themselves [123,124], and these changes are not reversed after transplantation into a young environment [124,125].

**Adult neural stem cells** (NSCs). Despite the fact that most neurons are postmitotic and have little potential for repair, there are several niches of NSCs in the adult brain that support neurogenesis in adulthood, i.e., the emergence of functionally active neurons de novo, from progenitor cells (radial glia, NSCs). Neurogenesis in the adult brain occurs throughout life in the subventricular (SVZ) and subgranular (SGZ) areas of the brain [for review, see [131,132]. The main functions of neurogenesis in the adult brain are the regeneration of olfactory cells and the formation of new neurons and gliocytes in structures that carry out the functioning of memory and other cognitive processes, and the maintenance of neuronal plasticity in general.

The number of NSCs, like many other resident stem cells of an adult organism, decreases with age, which, in turn, is accompanied by a decrease in the level of neurogenesis [133] and, as a consequence, a deterioration in cognitive functions. In contrast to other resident stem cells, however, the functions of “aged” NSCs are quickly normalized in vitro and are indistinguishable from those of the “young” NSCs [134]. Heterochronous parabiosis (connection of the circulatory systems of two animals of different ages) and restoration of Igf-1, Gh, Wnt3, Tgf-β or Gdf11 levels in old mice to the level of young mice significantly improves the level of neurogenesis in the former [135,136,137,138].

**Skin stem cells.** Skin is home to several types of stem cells including basal cells responsible for epithelial regeneration, hair follicle stem cells (HFSCs) mediating hair growth, and melanocyte stem cells that regenerate pigment-producing melanocytes. Hair follicles cycle between the phases of growth, regression, and rest (anagen, catagen, and telogen, respectively). The most pronounced age-related change in skin stem cells includes lengthening of, or in some cases, a complete shift to the resting period, which manifests as a lack of hair growth and loss of hair follicles (alopecia) [139]. Despite the natural hair loss during aging, the number of HFSCs does not decline [140,141]. Instead, there is a loss of their function, which underlies the lengthening of periods of rest. Unlike HFSC, the number of melanocyte stem cells in the skin decreases sharply with age. This decrease is not associated with apoptosis, but with ectopic differentiation and impaired self-renewal of stem cells [142]. Ionizing radiation and genotoxic stress contribute to melanocyte differentiation, which constitutes the main reason for the age-related hair greying [143]. 

**Germinal stem cells.** In mammals, male germline is maintained by spermatogonial stem cells (SSCs) that progressively become less numerous during aging [144,145]. Whereas males in mammalian species remain fertile throughout their lifetime, oogenesis in females of most mammals (except for several species of bats, bush babies, and chinchillas) stops before birth [146,147]. Some researchers have suggested that these species, and even possibly all other mammals, possess oogonial stem cells (OSC) capable of postnatally generating oocytes. OSCs have been isolated from mice and rhesus monkeys that generate oocytes in vitro and, in the case of mice, have been used to create transgenic pups [148,149]. OSCs have also been described in the adult human ovary. Such cells were demonstrated to produce oocyte-like cells upon transplantation into the ovarian tissue [150]. Yet, other research groups have failed to confirm post-natal oogenesis [151] or to isolate OSCs [152].

### 5.2. Basic Mechanisms of Stem Cell Aging

Despite the overall heterogeneity of aging process, there are several common themes in the functional decline observed in adult stem cells. Telomere shortening serves as a universal feature of aging. Even though telomerase expression is maintained in stem cell, the telomeres of HSCs, NSCs, HFSCs, and GSCs do shorten with age [153,154]. Telomere shortening was originally believed to be the main marker of cell and body aging. However, telomere end shortening was sometimes observed to poorly correlate with the chronological age [155]. Ake T. Lu et al. [156] researched DNA methylation in the telomeric and subtelomeric regions of chromosomes in peripheral blood leukocytes (formation of the DNAmTL tag), which is the most frequent cellular source for measurements of telomere length (TL). Their analysis indicated that DNA methylation is at least twice more accurate in predicting the age, compared to the TL quantification. Similarly, DNA methylation is much more accurate in assessing the impact of external negative factors (smoking, infections, etc) on the aging rate. Most interestingly, in in vitro experiments using hTERT-transduced cell lines, epigenetic age of the cells continued to grow despite the restored TL, and DNAmTL values for primary human fibroblasts decreased despite their hTERT-mediated immortalization. 

Accumulation of DNA damage and mutations is also known to contribute to the aging of stem cells. Medawar’s mutation accumulation hypothesis is one of the earliest theories explaining aging [157]. Phosphorylated histone H2aX in HSCs is a well-established marker of DNA damage, and is known to accumulate in HSCs and satellite cells during aging [129,158,159,160,161]. One of the suggested mechanisms of protection against DNA damage is that stem cells, such as HSCs and myosatellite cells, stay quiescent for long periods of time. This strategy indeed protects them from replicative stress, but at the same time renders them susceptible to accumulation of mutations. Upon DNA damage, double-stranded DNA breaks (DSBs) in resting cells will be repaired through error-prone non-homologous end-joining (NHEJ) rather than through homologous recombination (HR), which is far more accurate [162]. Thus, although proliferating stem cells are more likely to encounter DNA damage [163], they repair this damage more accurately than resting stem cells.

Stem cell self-renewal occurs during asymmetric cell division. This process can also be the key to the aging process, as it may contribute to the decline in self-renewal capacity via non-equal segregation of damaged DNA and protein molecules between daughter cells. One example of such asymmetric segregation is illustrated by the “immortal DNA strand hypothesis”, when parental DNA strand is invariably inherited by the daughter stem cell, whereas the newly synthesized complementary DNA strand that may contain some replication-induced mutations segregates into the differentiating daughter cell [164]. Likewise, damaged proteins and mitochondria are asymmetrically sorted by stem cells [165,166]. Each of those asymmetric divisions requires cell polarization. A decrease in the efficiency of SC polarization during asymmetric division with age contributes to aging and a decrease in the stem cell population. [167,168].

Metabolism also has a profound impact on the aging of stem cells, with periodic calorie deficit, the so-called calorie restriction (CR) having the largest effect. Short-term CR has been reported to increase the numbers of myosatellite cells [169] and to improve the functioning of various stem cell populations, including HSCs in mice [170] and GSCs in Drosophila [171]. CR also contributes to the ISC self-renewal in mice by enhancing expression of *bst1* in niche-forming Paneth cells. BST1 functions to convert NAD + into a paracrine signal of cyclic ADP-ribose (cADPR) sensed by the ISCs [172]. The activating effect of CR on stem cells can be carried out using various signaling systems, such as insulin-IGF, TOR, AMPK, sirtuins, and FOXO transcription factors [173].

At the molecular level in mammals, the effect of caloric restriction on aging retardation is transmitted through mTORC1 kinase, which balances catabolism and other processes [174]. Caloric restriction reduces activity which is characterized by increased lifespan in invertebrates and mice [175]. The circadian clock appears to negatively modulate the mTOR pathway by a mechanism that turns on BMAL1 [176]. In addition to diet, mTOR can also be pharmacologically inhibited by rampamycin and its analogs, slowing aging and providing a mild immunosuppressive effect to block chronic inflammation. [177,178,179].

The formation of damaging reactive oxygen species (ROS) and the activity of antioxidant systems play an important role in the metabolic mechanism of stem cell aging. In resting stem cells, the main mechanism for obtaining energy is glycolysis, which reduces the amount of ROS [180,181]. However, proliferating stem cells are very much dependent on the oxidative phosphorylation, which is accompanied with ROS production and oxidative stress. Altered expression of antioxidant defense components leads to their increased aging. In ISCs, this is associated with the formation of a cancer-prone environment [182]. Low mitochondrial activity in stem cells and progenitor cells may also contribute to age-dependent changes in stem cell functions [183].

All universal mechanisms of cell aging described above can be modulated by the transcriptional activity of circadian clocks as well as by the chromatin modifications underlying the activity of epigenetic clocks. Altered methylation patterns have been observed in aging HSCs. Specifically, genomic regions associated with open chromatin in lymphoid cells display higher DNA methylation in aging HSCs, whereas the opposite is observed for open chromatin regions in the myeloid cells [184]. H3K4me3, an activating histone modification, is known to become enriched during aging in the loci that control HSC self-renewal, which may potentially explain the age-dependent expansion of individual HSC clones [185]. Additionally, the levels of yet another activation-associated histone modification, H4K16ac, demonstrate an age-dependent decline in HSCs; inhibiting CDC42 restores H4K16ac to the level found in young HSCs and reverses the aging phenotype of transplanted HSCs [186]. In the myosatellite cells, H3K4me3 display moderate decline with age, whereas H3K27me3 levels significantly increase. Additionally, histone expression levels are reduced with age [187]. Expression levels of chromatin modifying enzymes such as the components of SWI-SNF and PRC complexes, HDACs (including sirtuins) and DNA methyltransferases have also been observed to change in the aging stem cells [105,188]. Taken together, these results indicate that changes in the epigenetic clocks are an integral universal feature of aging in stem cells, which determines their decline in function.

### 5.3. Reprogramming by Pluripotency Factors, Epigenetic Clock and Aging

Expression of four transcription factors, i.e., OCT4, SOX2, KLF4 and c-MYC (OSKM), converts somatic cells into induced pluripotent stem cells (IPSCs) [189]. Reprogramming occurs through global chromatin remodeling, which ultimately returns the cell to a pluripotent state corresponding to an embryonic stem cell (ESC), including the pattern of DNA methylation [for review, see 190]. This opens up great prospects for cell therapy; obtained autologous IPSCs can be differentiated into the desired cell type and thus “rejuvenate” cells, tissues and organs. However, cells with “zeroed” epigenetic age may inadequately respond to signals from the “adult” microenvironment, which, along with their genomic instability, may cause tumorigenesis [190,191,192].

Transcriptome analysis in single cell format revealed some interesting patterns in the process of obtaining IPSCs [193,194,195]. Thus, in the process of cellular reprogramming, two phases were distinguished. The first phase may be described as stochastic, characterized by differential expression of genes involved in the cell cycle (for example, *ccnb1* and *cdkn2b*), mesenchymal-epithelial transition (for example, Snai1 and Cdh1), as well as in the suppression of genes associated with cell adhesion and differentiation (e.g., Col1a1, Fbln5 and Mmp14) [194]. These initial changes are followed by a second phase, a deterministic or hierarchical one, characterized by progressive activation of pluripotency master genes (*nanog*, *oct4*, *sox2, c-Mic, dnmt3L* etc.) [194]. Epigenetic remodeling begins at the first phase of reprogramming and is characterized by the modification of histones such as H3K4me3 and H3K27me3, followed by changes in the DNA methylation profile, including those in the genes of *nanog*, *oct4* and *rex1*.

Olova N. et al. [196] showed that even with partial reprogramming, when the cell does not reach the state of pluripotency, age epigenetic signatures are zeroed out in the first 10 days of reprogramming, i.e., during the period of increased activity of pluripotency genes, and early expression of *NANOG*, *SALL4*, *ZFP42*, *TRA-1-60*, *UTF1*, *DPPA4* and *LEFTY2*. Similar results were demonstrated by Ocampo A. et al. [197], who performed incomplete cell reprogramming using cyclic OKSM induction. As a result, they showed a reduction in the epigenetic age to the early postnatal state without loss of cell specialization. Transient expression of only two transforming factors from the Yamanaka cocktail SOX2 and c-MYC was sufficient for the resulting cell population to be comparable in age signatures with the population differentiated from ESC [198]. Rejuvenation of reprogrammed cells has been demonstrated at the level of telomere measurement [199,200], mitochondrial rejuvenation, etc. [201,202,203]. Thus, partial reprogramming without reaching the pluripotent stage may be viewed as a method of epigenetic rejuvenation of cells and tissues.

Methods of reprogramming involving production of IPSCs in vivo in experiments on mice [204] theoretically make it possible to perform partial reprogramming with a “rejuvenating” effect in vivo. Thus, it has been shown that the cyclic expression of Yamanaki factors can increase the lifespan of progeria mice and improve cellular function in wild-type mice. [197]. An alternative approach to in vivo reprogramming has demonstrated the reversibility of age-related changes in retinal ganglion cells and the possibility of visual restoration in a mouse model of glaucoma [85]. More recently, it has been shown that transient non-integrative expression of nuclear reprogramming factors in vitro reverses many aspects of aging for human fibroblasts and chondrocytes, including resetting of epigenetic clock, reduction in the inflammatory profile, and restoration of youthful regenerative response [205].

A very recent publication by Gill D et al. available on bioRxiv claims to have developed an unprecedented transcription factor rejuvenation technology to rejuvenate human fibroblasts for as much as 30 years (as measured by a novel transcriptome clock) [206]. A technology called maturation phase transient reprogramming (MPTR) involves transfection of a polycistronic cassette with *OCT4*, *SOX2*, *KLF4*, *c-MYC*, and *GFP* genes under the control of a doxycycline promoter. Genes were expressed ectopically until maturation was reached, after which expression was turned off. MPTR has been shown to significantly rejuvenate all measured molecular signs of aging such as the epigenome, including H3K9me3 histone methylation levels and the DNA methylation aging clock without loss of cell identity.

During the development of non-integrating systems for cell reprogramming, it was found that hypoxia in the presence of FGF2 by activating the ESC-specific miR-302 cluster in primary and immortalized MSCs triggered the expression of the pluripotency genes *OCT4* and *NANOG* in the immortalized L87 cell line and primary MSCs, which was accompanied by an increase in the rate proliferation and inhibition of aging [207]. Hypoxia also improves MSC reprogramming due to episomal expression of pluripotency factors. Interestingly, the sirtuins mentioned in connection with the function of the circadian clock are also involved in the process of cell reprogramming in the pluripotent direction [208]. SIRT1 can deacetylate the SOX2 transcription factor and regulate the reprogramming of IPSCs through the miR-34a-SIRT1-p53 axis. SIRT2 regulates the function of IPSCs through GSK3β, and SIRT3 can positively regulate the expression of PPAR 1-alpha gamma coactivator (PGC-1α) during stem cell differentiation. SIRT5 deacetylates STAT3, which is an important transcription factor in the regulation of stem cell pluripotency and differentiation. SIRT6 can enhance the reprogramming efficiency of IPSCs from old skin fibroblasts via miR-766 and increase the expression levels of reprogramming genes, including *SOX2*, *OCT4*, and *NANOG*, by acetylation of histone H3 lysine. SIRT7 plays a regulatory role in the mesenchymal-epithelial transition, which is thought to be a critical process in the generation of IPSCs from fibroblasts [208]. Thus, the above examples again emphasize the relationship of genes that determine the function of circadian rhythms, epigenetic clocks and regulation of stem cell aging.

### 5.4. Epigenetic Approaches to Stem Cell Rejuvenation

While full and partial reprogramming using transcription factors of pluripotency can be characterized as a direct effect on chromatin as a substrate for the epigenetic clock, other approaches to rejuvenation associated with the activation of certain signaling pathways affect the epigenetic clock indirectly, i.e., through circadian clock mechanisms. 

Let us consider approaches to stem cell rejuvenation using the example of MSCs as one of the most studied types of stem cells in regenerative medicine. Specific interest in MSCs is due to the fact that, along with the availability and relative simplicity of propagation [108], MSCs have significant paracrine therapeutic potential, which, consequently, has led to their widespread clinical use [209,210]. The effectiveness of MSCs has been shown in the treatment of various diseases, including graft versus host disease [211], Crohn’s disease [212], diabetes mellitus [213], multiple sclerosis [214], myocardial infarction [215] etc. As we have already noted, the functional activity of MSCs significantly decreases with age. In an attempt to “rejuvenate” MSCs, various genetic modifications were undertaken, processing of microRNAs and noncoding RNAs [216], preconditioning under hypoxic conditions or in the presence of various cytokines, etc. [217,218]. Many researchers use MSCs as an object of partial reprogramming by pluripotency factors, as described in the previous section. [218,219,220,221].

Liu G.H. et al. [222] showed that CLOCK deficiency accelerates the aging of hMSC, while overexpression of CLOCK, even in a transcriptionally inactive form, rejuvenates old hMSC. Based on the idea that CLOCK forms complexes with nuclear membrane proteins and KAP1 and thus stabilizes heterochromatin, the authors, by editing the CRISPR/Cas9 genome, increased the expression level of CLOCK in senescent MSCs, which led to their rejuvenation and promoted cartilage regeneration in mice.

In a recent study by Jiao H, a global transcriptome analysis of conventional MSCs and rejuvenated MSCs performed through the pluripotency stage was performed [223]. Reduced level of expression of *GATA-binding protein 6* (*GATA6*), the expression of which is associated with the basal-like chemoresistant subtype of pancreatic ductal adenocarcinoma was found in the latter ones [224,225]. The expression level of *GATA6* is inversely correlated with the activity of the SHH (sonic hedgehog) signaling pathway and the expression level of forkhead box P1 (FOXP1). FOXP1 is known to slow aging by directly regulating p16INK4A transcription in MSCs. [226]. Thus, *GATA6* knockout can be used to rejuvenate MSCs ex vivo.

Recently Deng et al. obtained an experimental model of DiGeorge syndrome on MSCs with a deficiency of the critical region 8 (DGCR8) with an accelerated aging phenotype. Mechanically, DGCR8 supports the organization of heterochromatin by interacting with the nuclear envelope protein Lamin B1 and the heterochromatin-associated KRAB-associated protein 1 (KAP1) and heterochromatin protein 1 (HP1), thus regulating the aging of MSCs [227]. Likewise, yes-associated protein (YAP) was first identified as a major effector of Hippo signaling that plays an important role in cell development and differentiation. Fu et al. generated YAP-deficient MSCs with a premature cell senescence phenotype and found that YAP cooperates with the TEM domain transcription factor (TEAD) to activate *FOXD1* [226].

Ren et al. generated CBX4-deficient MSCs and demonstrated that CBX4 counteracts aging of MSCs by maintaining nucleolar homeostasis by recruiting the nuclear protein fibrillarin and the heterochromatin component KAP1 to the nucleolar rDNA to stabilize it, thereby limiting rRNA overexpression and slowing down cellular senescence [228]. Gene therapy using lentiviral transduction of the *DGCR8* gene effectively attenuates the aging of MSCs, as evidenced by the suppression of markers of cell aging (p16^INK4a^ and CDKN1A) and inflammation factors [227]. By analogy, lentivirus-mediated *YAP*, *FOXD1*, or *CBX4* gene transfer also rejuvenates old MSCs [226,228].

Several studies have focused on the epigenetic modulation of senescent MSCs. Thus, gene expression can be regulated by DNA methylation by suppressing the activity of the corresponding promoter regions. 5-Azacytidine (5-AZA), a DNA methyltransferase (DNMT) inhibitor, reverses the aging MSC phenotype by reducing the accumulation of ROS (reactive oxygen species), improving superoxide dismutase activity and increasing the BCL-2/BAX ratio [229]. The RG108 DNA methyltransferase inhibitor significantly induces TERT expression by blocking methylation in the TERT promoter region. DNMT1 and DNMT3B belong to methyltransferases that modulate polycomb-mediated histone methylation patterns, which are significantly reduced during replicative senescence of MSCs. DNMT3a expression was found to increase during replicative senescence, which correlates with CPG hypermethylation in old MSCs. [230]. In this regard, it is assumed that demethylation in promoter regions plays an important role in maintaining the MSC phenotype, lengthening the life cycle, and regeneration. It has also been demonstrated that tetramethylpyrazine (TMP) significantly slows down cell aging by modulating EZH2 (the histone lysine N-methyltransferase enzyme) -H3k27me3, which suggests trimethylation at the 27th lysine residue of the histone protein H3 [231]. Restoration of mitochondrial NAD + levels through overexpression of *NNT* and *NMNAT3* and delayed replicative senescence can increase the efficiency of reprogramming old MSCs [232].

Several studies have shown that aging of MSCs can be reversed by modulating ROS aggregation and oxidative metabolism. Ascorbic acid has been shown to inhibit D-galactose-mediated ROS production and activation of AKT/mTOR signaling in MSCs [233]. Another study showed that lactoferrin inhibits hydrogen peroxide-induced ROS production and suppresses caspase-3 and AKT activation to reduce hydrogen peroxide-induced apoptosis [234]. MSCs pretreated with Cirsium setidens, a kind of antioxidant, can inhibit ROS production and reduce the expression of phosphorylated mitogen-activated protein kinase p38, N-terminal kinase c-Jun and p53 [235]. The combination of mitochondrial biogenesis, mitochondrial dynamics, and mitophagy determines mitochondrial morphology and mitochondrial function. Mitochondrial dysfunction is often viewed as a typical phenotype of senescent cells. For example, melatonin can slow the aging of MSCs by enhancing mitophagy and mitochondrial function by activating the 70 kDa heat shock protein 1L (HSPA1L) [236]. HSPA1L binds to COX4IA, a mitochondrial complex IV protein, resulting in increased mitochondrial membrane potential and antioxidant enzyme activity [236]. Decreased CPT1A (carnitine palmitoyltransferase 1A) reverses mitochondrial dysfunction and aging of MSCs [237]. It has also been shown that increasing FGF21 levels can improve mitochondrial function, rejuvenating aging MSCs, and regulating mitochondrial dynamics [238]. The work of mTORC1 also regulates autophagy, whose the decline in activity is observed with aging. The inhibition of mTORC1 by AICAR and NAM enhances autophagy and preserves the ability of cells to self-renew and differentiate, as well as postpones aging-related changes [239].

In the previous sections, we talked a lot about the role of sirtuins in circadian clock control, metabolism, and aging. Sirtuins emerged as global metabolic regulators that control the response to calorie restriction and defense against age-related diseases, thus extending both the healthy period and, in some cases, life expectancy [240,241]. Sirtuin-activating compounds (STACs) increase sirtuin activity and increase lifespan in mice and nonhuman primates [242]. Liu B.’s group [243] showed in an experiment on mice that ectopic expression of *sirt7* relieves the inflammatory response caused by progerin in endothelial cells, which is characteristic of aging and Hutchinson-Guildford syndrome (HGPS). *sirt7* gene therapy directed at the vascular endothelium, driven by the *icam2* promoter, improves neovascularization, decreases aging rates and increases lifespan in Lmna ^f/f^ mutant mice.

Several studies in regenerative medicine focus on the use of exosomes containing various biologically active substances, both from MSCs and from other types of SC. In a number of studies, exosomes have been used to rejuvenate cells. Thus, extracellular vesicles obtained from embryonic stem cells (ESC-Exos) were used as a factor of MSC rejuvenation mediated by the IGF1/PI3K/AKT signaling pathway [244]. Inhibition of PI3K/AKT/mTOR significantly increases the expression of some pluripotency genes such as *NANOG* and *OCT4* [245]. Many studies have shown that NANOG effectively reverses the aging of MSCs [246]. Various underlying mechanisms have been proposed. NANOG activates *PBX1* (homeodomain transcription factor) and activates the AKT signaling pathway. A feedback loop likely exists between PBX1 and AKT signaling, keeping MSCs in a highly proliferative state with good differentiation potential [245]. NANOG also restores *COL3* expression and thus stabilizes extracellular matrix synthesis [247]. The use of ESC-Exos has also been shown to rejuvenate endothelial cells, enhance angiogenesis, reduce ROS levels, and promote pressure ulcer regeneration in aging mice [248,249]. The use of exosomes, in contrast to reprogrammed cells, is not accompanied by any oncological risks and therefore seems to be a very promising and interesting direction in antiaging.

## 6. Conclusions

Aging is a complex genetically determined process, the implementation of which has its own characteristics both at the cellular level in resting, proliferating, and differentiating stem cells, and at the organ and organism levels. Like a grand orchestra, the mammalian organism, for the synchronization of aging processes and the adequate inclusion of genetic programs in response to external aging factors in cells, organs and tissues must depend on a common conductor. Such a common conductor may be found in the circadian clock regulated by the central oscillator in the hypothalamus and the interrelated genetic mechanisms of DNA methylation regulation, which form the basis of the epigenetic clock, with the help of which the biological age of cells, tissues and organs is determined. The close relationship of aging, regeneration and oncogenesis with each other and with the central clockwork requires further precision research, and we are certain that the result of such research may manifest itself in the creation of an effective strategy for antiaging as well as prolonging the active life of humans.

## Figures and Tables

**Table 1 jpm-11-01050-t001:** Basic Molecular Factors Controlling Circadian Rhythms.

Basic Molecular Factors	Role in Control	Reference
Basic Molecular Factors Controlling Circadian Rhythms		
CLOCK	Central proteins of the mammalian clockworks; serve as the major transcription activators of b-HLH-PAS (The basic helix–loop–helix/Per-ARNT-SIM) expression, which in turn modulates the transcription of multiple clock-controlled genes (CCGs).	[10]
BMAL1	Central proteins of the mammalian clockworks; serve as the major transcription activators of b-HLH-PAS (The basic helix–loop–helix/Per-ARNT-SIM) expression, which in turn modulates the transcription of multiple clock-controlled genes (CCGs).	[10]
PER	Forms a transcription-translation negative feedback loop for CLOCK/ BMAL1 complex. Translated from the three mammalian homologs of drosophila-per, one of three PER proteins (PER1, PER2, and PER3) dimerizes via its PAS domain with one of two cryptochrome proteins (CRY1 and CRY2) to form a negative element of the clock. This PER/CRY complex moves into the nucleus upon phosphorylation by CK1-epsilon (casein kinase 1 epsilon) and inhibits the CLOCK/ BMAL1 heterodimer, the transcription factor that is bound to the E-boxes of the three per and two cry promoters by basic helix-loop-helix (BHLH) DNA-binding domains.	[35,59,68]
CRY	Cryptochrome; one of the four groups of mammalian clock genes/proteins that generate a transcription-translation negative-feedback loop (TTFL), along with Period (PER), CLOCK, and BMAL1. In this loop, CLOCK and BMAL1 proteins are transcriptional activators which bind together to the promoters of the *Cry* and *Per* genes and activate their transcription. The CRY and PER proteins then bind to each other, enter the nucleus, and inhibit CLOCK-BMAL1-activated transcription.	[35]
REV-ERBα/β	Occupies the promoter elements of Reb-Erb/ROR. Rev-Erbα plays an important role in regulation of the core circadian clock through repression of the positive clock element Bmal1.The secondary TTFL, featuring Rev-Erbα working in conjunction with Rev-Erbβ and the orphan receptor RORα, is thought to strengthen this primary TTFL by further regulating BMAL1.	[36]
RORα	Occupies the promoter elements of Reb-Erb/ROR. RORα shares the same response elements as Rev-Erbα but exerts opposite effects on gene transcription; BMAL1 expression is repressed by Rev-Erbα and activated by RORα.	[36]
p300	Functions as the co-activator of CLOCK-BMAL1 and launches the cyclic acetylation of histone tails.	[43,44,45]
CBP	Operates similarly to p300, and has been described to interact with PER2 to repress CLOCK-BMAL1 activity.	[43,44,45]
MLL1	Tri-methylates histone H3K4 on the nucleosomes near the circadian gene promoters, which results in transcription activation.	[46,47]
Jarid1A	A major component of the circadian clock, the upregulation of which at the end of the sleep phase blocks HDAC1 activity. Blocking HDAC1 activity results in an upregulation of CLOCK and BMAL1 and consequent upregulation of PER proteins. The PSF (polypyrimidine tract-binding protein-associated splicing factor) within the PER complex recruits SIN3A, a scaffold for assembly of transcriptional inhibitory complexes, and rhythmically delivers histone deacetylases to the Per1 promoter, which repress Per1 transcription.	[47]
EZH2	The catalytic subunit of the Polycomb Repressive Complex 2 (PRC2). As a histone methyltransferase (HMTase), EZH2’s primary function is to methylate Lys-27 on histone 3 (H3K27me) by transferring a methyl group from the cofactor S-adenosyl-L-methionine (SAM).	[48]
Jmjd5	Contains a jumonji-C (jmjC) domain that is often found in proteins with histone demethylase activity; indeed, its human ortholog (also known as *KDM8*) has recently been shown to have this function. Jmjd5 is coregulated with evening-phased clock components and positively affects expression of clock genes expressed at dawn.	[49]
SIRT1	Was found to deacetylate a number of proteins downstream of Myc, p53, HIF, TGF-β, and Wnt signaling, and acts as a tumor suppressor or a tumor-promoting factor, depending on the specific biological context. Also, SIRT1 is involved in the adaptation of the organism to nutritional starvation. SIRT1 is required for VMH and ARC neurons to control glucose metabolism and lipid exchange on the periphery; deregulation of these processes is associated with aging. In the context of SCN, SIRT1 modulates the activity of the central pacemaker.	[51,52,53,54,55,56,57,58,62]
SIRT6	Is mainly known as a deacetylase of histones H3 and H4, an activity by which it changes chromatin density and regulates gene expression. The enzymatic activity of Sirt6, as well as of the other members of the sirtuins family, is dependent upon the binding of the cofactor nicotinamide adenine dinucleotide (NAD^+^).	[52]
SIRT3	Is believed to set the acetylation rhythms as well as OXPHOS oscillations in liver mitochondria. Overexpression of *SIRT3* in cultured cells increases respiration and decreases the production of reactive oxygen species. Activation of the NMNAT2 enzyme, which catalyzes an essential step in the nicotinamide adenine dinucleotide (NAD+) biosynthetic pathway by SIRT3, may be a means of inhibiting axon degeneration and dysfunction. Activation of SIRT3 inhibits the apoptosis leading to age-related macular degeneration.	[60]

## Data Availability

Not applicable.

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
