# Peer review of "Epigenetic Clock and Circadian Rhythms in Stem Cell Aging and Rejuvenation"

_jpm, 2021, doi:10.3390/jpm11111050_

Round 1

Reviewer 1 Report

The authors desribe interactions of the circadian clock and the epigenetic clock in the context of both cellular and oragnism aging. They provide a thourough use up-to-date literature to describe each of the components, underline examples of their potential interaction, and give insight in applicability of this knowledge in stem cell research. The manuscript is well written, in a comprehensible manner, and suggests new possibilites for the fields of aging as well as circadian research. I have no issues for consideration.

Author Response

We are grateful to the Reviewer fot the high estimation of our work.

Reviewer 2 Report

The title of the review and abstract clearly set the tone for an exciting manuscript, providing clear concepts on interaction between circadian rhythms of gene expression and epigenetic clocks during aging. Unfortunately, however, the content of the review is far from being clear because the authors put too much information on stem cell aging. A number of structural changes (including the omission of a large number of statements and information in the chapter 5, and in contrast expand the contents for the chapter 4) are required, in order to render the review suitable for publication.

  1. The first half of the manuscript are informative, well-organized, centered on two main clock mechanisms. However, second half about cell aging are less-organized and some information about the tissue stem cells are even not related to epigenetic clock or circadian rhythms and can be excluded from the manuscript.
  2. Too much spaces are about stem cells aging and reprogramming, which is far from the readers’ anticipation for the current title of ‘Epigenetic Clock, Circadian Rhythms and Cell Aging’. Therefore the title/abstract are better to be modified. Or the latter can be summarized in half portion.
  3. Throughout the whole manuscript, neither figures nor tables were used. It would be helpful if the roles of Sirt family members or histone modifications on cell aging were summarize in either format. 

Author Response

We are grateful to the Reviewer for the valuable comments which helped us to improve the quality of the manuscript.

In accordance with Reviewer’s suggestions, we modified the title and the abstract by adding passages about stem cell aging and rejuvenation. The idea of this review was to investigate the interactions between circadian rhythms, epigenetic clocks and stem cell aging. We believe that the last sections of the review are the most interesting in terms of creating cellular rejuvenation technologies using circadian and epigenetic clocks to assess the results. We also shortened the second half of the review and added the Table 1, which summarized the data about the main actors of circadian clocks. All insertions in the text of the article are marked with blue font.